# Malian Children’s Core Gut Mycobiome

**DOI:** 10.3390/microorganisms12050926

**Published:** 2024-05-01

**Authors:** Abdourahim Abdillah, Aly Kodio, Stéphane Ranque

**Affiliations:** 1IHU-Méditerranée Infection, 13385 Marseille, France; abdourahim15@live.fr (A.A.); alkodio@icermali.org (A.K.); 2Malaria Research and Training Centre-International Center for Excellence in Research (MRTC-ICER), Department of Epidemiology of Parasitic Diseases, Faculty of Medicine and Dentistry, Université des Sciences des Techniques et des Technologies de Bamako, Point G, Bamako BP 1805, Mali; 3AP-HM, RITMES, Aix-Marseille Université, 13005 Marseille, France

**Keywords:** fungal gut community structure, gut mycobiome, core mycobiome, ITS metabarcoding, children, Dogon country

## Abstract

Because data on the fungal gut community structure of African children are scarce, we aimed to describe it by reanalysing rRNA ITS1 and ITS2 metabarcoding data from a study designed to assess the influence of microbiota in malaria susceptibility in Malian children from the Dogon country. More specifically, we aimed to establish the core gut mycobiome and compare the gut fungal community structure of breastfed children, aged 0–2 years, with other age groups. Briefly, DNA was extracted from 296 children’s stool samples. Both rRNA ITS1 and ITS2 genomic barcodes were amplified and subjected to Illumina MiSeq sequencing. The ITS2 barcode generated 1,975,320 reads and 532 operational taxonomic units (OTUs), while the ITS1 barcode generated 647,816 reads and 532 OTUs. The alpha diversity was significantly higher by using the ITS1 compared to the ITS2 barcode (*p* < 0.05); but, regardless of the ITS barcode, we found no significant difference between breastfed children, aged 0–2 years, compared to the other age groups. The core gut mycobiome of the Malian children included *Saccharomyces cerevisiae*, *Candida albicans*, *Pichia kudriavzevii*, *Malassezia restricta*, *Candida tropicalis* and *Aspergillus* section *Aspergillus*, which were present in at least 50% of the 296 children. Further studies in other African countries are warranted to reach a global view of African children’s core gut mycobiome.

## 1. Introduction

The human digestive tract is a complex polymicrobial ecosystem composed of several microorganisms such as fungi, bacteria and viruses, which colonize the host throughout life. In recent years, with the emergence of new sequencing technologies, characterization of the gut mycobiome has become a major issue. Indeed, fungal dysbiosis has been shown to be associated with certain diseases including colorectal cancer and inflammatory bowel disease [1,2,3]. Most studies of the healthy gut mycobiome have been performed on adult subjects living in industrialized countries. In this population, the mycobiome profile seems to be characterized by a dominance of the genera *Saccharomyces*, *Malassezia* and *Candida* [4,5]. Since microbial gut dysbiosis in early life has been shown to impact children’s development [6], studying the gut mycobiome might help to characterize specific profiles associated with certain diseases. However, data on the gut mycobiome in healthy children are scarce. The few existing studies have been conducted on children from industrialized countries in Europe and America. In this population, the mycobiome profile has been characterized by a dominance of the genera *Penicillium*, *Aspergillus*, *Candida* and *Saccharomyces* [6,7]. In non-industrialized countries, particularly in Africa, little is known regarding the gut mycobiome profile of children. Recently, *Candida*, *Malassezia* and *Aspergillus* were found to be dominant in the gastrointestinal tract of 0–5 year-old children of a rural community in Ghana [8]. Whether this is common to all African children remains unknown. To address this knowledge gap, the present study aimed to describe the fungal gut community structure of African children by reanalysing ITS metabarcoding data from a study on the influence of microbiota in malaria susceptibility in Malian children [9]. The objectives of this ancillary study were: (1) to determine the core gut mycobiome of Malian children, and (2) to examine the effect of age by comparing breastfed children aged 0–2 years with other age groups via ITS metabarcoding.

## 2. Patients and Methods

### 2.1. Study Participants and Stool Collection

A longitudinal cohort study was conducted from October 2017 to December 2018 at the Bandiagara Malaria Project (BMP) clinical research centre in Mali, as previously described [10]. The study involved children aged from six months to years who were not taking drugs with known antimalarial activity or antibiotics and who had no clinical symptoms of disease. Children’s faeces were collected at day 0 in identified sterile jars and immediately placed at 4 °C. Hard stools were diluted *v*/*v* with 10X PBS (Phosphate-Buffered Saline pH 7.4, RNase-free; Thermo Fisher Scientific, Bourgoin-Jallieu, France) solution. Stool aliquots were distributed into 1 mL tubes, kept at 20 °C in Mali, then packed in dry ice and shipped to Marseille for further metabarcoding studies.

### 2.2. Stool DNA Extraction

DNA was extracted using the semi-automatic extraction protocol of EZ1 Advanced XL (QIAGEN Instruments Hombrechtikon, Switzerland) with the EZ1 DNA tissue kit (Qiagen GmbH, Hilden, Germany) as described herein [10]. 

### 2.3. ITS Metabarcoding

We used both the ITS1 and ITS barcode and two hybridization temperatures in triplicates to maximize the diversity of the fungal reads. All the amplification procedures and the steps of the Illumina MiSeq sequencing until the production of the reads have been previously described [9]. Briefly, the amplification reaction mix consisted of 12.5 µL AmpliTaq Gold master mix, 0.75 µL of each primer with Illumina sequencing adapters (Eurogentec, Seraing, Belgium), 6 µL distilled water and 5 µL DNA template for 25 µL volume. The amplification programme was as follows: 95 °C for 10 min; 45 cycles of 95 °C for 30 s, (55 °C or 52 °C) for 30 s, and 72 °C for one minute; and 72 °C for five minutes. Amplifications of the ITS1 and ITS2 regions using the primers described herein [10] with independent hybridization temperatures of 52 °C and 55 °C were made in triplicate. The amplicons of the replicated PCRs and the two hybridization temperatures relating to ITS1 and ITS2 were pooled for metabarcoding on the MiSeq platform.

After purification on AMPure beads (Beckman Coulter Inc., Fullerton, CA, USA), concentration was measured using high sensitivity Qubit Flex technology (Beckman Coulter Inc., Fullerton, CA, USA) and dilution to 3.5 ng/µL was performed. At this step, dual-index barcodes were added to the amplicon. After purification on AMPure beads (Beckman Coulter Inc., Fullerton, CA, USA), this library was pooled with 94 other multiplexed samples. The global concentration was quantified by a Qubit assay with the high-sensitivity kit (Life Technologies, Carlsbad, CA, USA). Before loading for sequencing on MiSeq (Illumina Inc., San Diego, CA, USA) the pool was diluted at 8 pM. Automated cluster generation and paired-end sequencing with dual-index reads were performed in a single 39 h run in a 2 × 250 bp. The paired reads were filtered according to the read qualities. The raw data were configured in fastq files for R1 and R2 reads.

The Illumina MiSeq sequences analysis was performed by PIPITS, an automated pipeline for the analysis of fungal ITS (internal transcribed spacer) sequences from the Illumina sequencing platform, hereafter referred to as the protocol [11]. The pipeline consists of the following consecutive steps: (1) preparation of raw sequences (joining, conversion, quality filtering, relabelling and file formatting), (2) extraction of ITS fungi and read reorientation, and (3) processing of the reads to produce an operational taxonomy unit (OTU) abundance table and taxonomic assignment table for downstream analysis. In this case, the extracted ITS2 and ITS1 sequences were analysed for the processes to obtain the OTU table. The OTU sequences were defined as a cluster of 97% sequence identity. The last step generated the repseqs.fasta file representing the OTU sequences. These OTU sequences were manually queried via BLASTN against the nucleotide NCBI with the search parameters: (1) rRNA genes internal transcribed spacer region (ITS) from fungi type and reference material and, if the first query yielded <97% identity, (2) the nucleotide collection (nt) to improve the taxonomic assignment that was generated by PIPITS. The taxon selection criteria were defined as follows: PID (percentage of identity) > 97% assignment to species; PID between 95 and 97%: assignment to genus level; PID between 90 and 95%: assignment to the family; PID below 90%: assignment to the kingdom.

### 2.4. Statistical Analysis

We used the R software version 4.0.2 (www.R-project.org/, accessed 15 June 2021 [12]) to compute the alpha and beta diversity. Alpha diversity was estimated by the observed richness and Shannon diversity. Boxplots were made using the *ggplot2* R package. To test whether there are differences in diversity between the amplified ITS regions, we used analysis of variance ANOVA to compare ITS1 and ITS2 using the *rstatix* R package. To test for differences in diversity between ages, ANOVA was also applied between age groups. A *p*-value lesser than 0.05 was considered statistically significant.

### 2.5. Ethics and Consent of Participants

This study was approved by the Ethics Committee of the Faculty of Medicine of Mali (No 2017/133/CE/FMPOS). Written informed consent was obtained from each child and at least one parent or legally responsible adult.

## 3. Results

### 3.1. Demographic Characteristics of the Children

A total of 300 healthy children were included at the Bandiagara Malaria Project (BMP) clinical research centre in Bandiagara, Mali, as previously defined [10]. We collected one stool sample from each child and four samples were excluded, giving a total of 296 samples. Of the 296 children studied (average age, 7.5 ± 3.9), there were 143 males and 153 females. We classified the children into 3 age groups of 0–2 years (mean age 1.2 ± 0.7, *n* = 37), 3–8 years (mean age 5.5 ± 1.7, *n* = 130) and 9–15 years (mean age 11.4 ± 1.7, *n* = 129) to determine their gut mycobiome. 

### 3.2. ITS Metabarcoding and Fungal Diversity

ITS sequencing produced a total of 2,623,136 reads, representing 1011 single operational taxonomic units (OTUs) for the two amplified regions. ITS2 generated 1,975,320 reads and 532 OTUs while ITS1 generated 647,816 reads and 532 OTUs. For further analysis, we grouped together OTUs that designated the same species. We used the observed species richness and Shannon diversity to measure alpha diversity. When comparing the amplified ITS regions, alpha diversity was significantly higher in ITS1 compared to ITS2 (*p* < 0.05; Figure 1A). However, there was no difference in alpha diversity between the 3 age groups regardless of the amplified ITS region (*p* > 0.05, Figure 1B,C).

### 3.3. Relative Abundance and Prevalence

We analysed the dominant phyla and the most prevalent and abundant genera of the gut mycobiome of the children. Ascomycota and Basidiomycota were the most abundant phyla with 71.6% and 26.4% in ITS1 and 93.5% and 6.4% in ITS2, respectively (Figure 2A,B). At the genus level, four genera were most abundant in ITS1, including *Pichia*, *Aspergillus*, *Candida* and *Saccharomyces* while in ITS2, *Saccharomyces*, *Aspergillus*, *Malassezia*, *Candida* and *Pichia* were most abundant (Figure 2C,D). By analysing the genera present in at least 50% of all children, *Pichia* (90.9%), *Aspergillus* (88.5%), *Candida* (81.1%) and *Saccharomyces* (76%) were the most prevalent in ITS1 while *Saccharomyces* (99.6%), *Malassezia* (80.4%), *Candida* (76.3%), *Pichia* (75.7%) and *Aspergillus* (59%) were the most prevalent in ITS2 (Table 1). The prevalence of the genera among the three groups of children was quite homogeneous in ITS1 and ITS2 (Figure 3A,B). At the species level, *Pichia kudriavzevii* (90.9%), *Aspergillus* section *Aspergillus* (66.9%), *Candida tropicalis* (55.7%), *Candida albicans* (53.4%) and *Saccharomyces cerevisiae* (50.7%) were the most prevalent in ITS1, while *Saccharomyces kudriavzevii* (99.3%), *Saccharomyces cerevisiae* (91.2%), *Malassezia restricta* (78.7%), *Pichia kudriavzevii* (75.3%) and *Candida albicans* (64.5%) were the most prevalent in ITS2 (Table 1). The prevalence of species among the three groups of children was also quite homogeneous in ITS1 and ITS2, showing that there is no difference (Figure 3C,D).

### 3.4. Fungal Species Shared between Children

To determine the gut mycobiome shared among the children, we drew Venn diagrams of the fungal species detected in ITS1 and ITS2. In ITS1, 265 distinct fungal species were detected. Among them, 12 species were detected only in children aged 0–2 years, 52 species in 3–8 years and 32 in 9–15 years (Figure 4A). In ITS2, 217 distinct fungal species were detected and among them, 10 species were detected only in children aged 0–2 years, 50 species in 3–8 years and 40 in 9–15 years (Figure 4B). In both ITS1 and ITS2, the species detected in each group were present at a very low prevalence and cannot be considered age-related. Comparing the common mycobiome of the three groups of children, 113 species were detected in ITS1 and 73 species were detected in ITS2 (Figure 4A,B). When analysing the shared ITS1 and ITS2 mycobiome, we found 26 fungal species amplified by both regions (Figure 4C). Among them, six fungal species including *S. cerevisiae*, *C. albicans*, *P. kudriavzevii*, *M. restricta*, *C. tropicalis* and *Aspergillus* section *Aspergillus* were present in at least 50% of all children, showing that they represent the core gut mycobiome of Malian children. Interestingly, we observed that some species were amplified by a single ITS region with a highly increased prevalence. This was the case for *Wallemia tropicalis* (41.2%), which was exclusively detected with ITS1, and *Saccharomyces kudriavzevii*, which was almost exclusively (99.3%) detected with ITS2. This highlights an important bias in the choice of the amplified target in fungal metagenomic studies (Figure 4C). 

## 4. Discussion

Fungi in the human gut may result from passage or colonization through the digestive tract from the first days of life. Here, we studied gut mycobiome structure in Malian children. We found no significant difference in alpha diversity between breastfed children aged 0–2 years with those aged 3–8 years and 9–15 years. These results are in line with those of studies comparing healthy Italian children aged 0–2 years, 3–10 years and 11–17 years [7], and 0–5 years-old children from a rural community in Ghana [8]. This suggests a relatively stable fungal diversity from birth to adolescence. Regarding the fungal genera structure, we found that *Pichia*, *Candida*, *Saccharomyces*, *Aspergillus* and *Malassezia* were the most abundant in the children’s gut mycobiomes, with slight variations among the three age groups. These findings are in line with those of children from a rural community in Ghana where *Candida*, *Malassezia* and *Aspergillus* were found to be the most dominant [8]. In contrast, *Penicillium*, *Aspergillus* and *Candida* were found to be most abundant in healthy Italian children of the same age group [7]. One explanation for the high abundance of *Malassezia*, *Candida*, *Aspergillus*, *Saccharomyces* and *Penicillium* in the digestive tract of children is that these genera have been found in the human breast milk mycobiome [13,14].

At the species level, we found that *Pichia kudriavzevii*, *Aspergillus* sect. *Aspergillus*, *C. tropicalis*, *C. albicans*, *S. cerevisiae*, *S. kudriavzevii* and *M. restricta* were the most prevalent. In rural Ecuadorian children from 3 months of age, a fungal dysbiosis, characterized by a high abundance of *P. kudriavzevii*, was associated with childhood atopic wheeze to 5 years of age [6]. The high prevalence of *P. kudriavzevii* observed in our studies, therefore, deserves further investigation. By analysing the gut mycobiome shared by the children, we found 103 and 69 species by using ITS1 or ITS2 barcoding, respectively. However, 31 species were found and amplified by both ITS1 and ITS2. Among them, *S. cerevisiae*, *C. albicans*, *P. kudriavzevii*, *M. restricta*, *C. tropicalis* and *A. sect Aspergillus* were detected in at least 50% of all children, suggesting that they constitute the core gut mycobiome of Malian children. *S. cerevisiae*, *C. albicans* and *M. restricta* were shown to belong to the core gut mycobiome in healthy adults [5]. These data suggest that there is a stable colonization of the human gut with some fungal species since birth. However, here we come up against the major limitation of the metagenomic approach, which neither differentiates living and dead DNA nor fungi that truly colonize the intestine and those that are simply in transit, mainly brought in by food. In this respect, our findings are currently too fractional to support further hypotheses on the function and/or ecological significance of these fungi.

Our analysis of the Malian children’s gut mycobiome structure showed that the alpha diversity was significantly higher by amplifying the ITS1 region compared to the ITS2, but we found no significant difference between breastfed children aged 0–2 years compared to other age groups. 

Fungal metagenomics studies show heterogeneity in the choice of the amplified barcode. Some studies used ITS1 [7,15], while many have preferred using ITS2 [5,8,16,17,18] as a barcode for fungal identification. Whereas the barcode is a key determinant of the results, it is rarely discussed in the studies’ final results. Using both ITS1 and ITS2 in two studies of the gut mycobiome, we have previously found that there are biases in amplification in both regions [19,20]. The relatively higher alpha diversity that we observed by using the ITS1 barcode compared to the ITS2 barcode in this study is in agreement with others who compared the performance of the ITS1 and ITS2 barcodes in an environmental study [21]. Moreover, we found disparities in abundance and prevalence at the phyla, genus and species level when comparing the ITS1 or ITS2 barcode results. For a given species, both abundance and prevalence varied by using the ITS1 or the ITS2 barcode. For example, *P. kudriavzevii* abundance and prevalence were 21.8% and 90.9% by using ITS1, compared to 10.5% and 75.3% by using the ITS2 barcode. The most striking results were the amplification of some species by only ITS1 or ITS2 with a particularly high prevalence, which clearly shows the amplification bias in fungal metagenomic studies. Similar disparities in ITS1 and ITS2 have also been highlighted by others [20,21]. 

The main strengths of our study were the relatively large sample size and the analysis of the mycobiome using both ITS1 and ITS2 barcodes. The limitations of our study include the lack of follow-up over time, which could assess the stability of the digestive mycobiome, and the lack of information on the children’s diet, which might explain the abundance of certain fungal species in the children’s gut mycobiome.

In conclusion, the core gut mycobiome of Malian children includes fungal species of the genera *Candida*, *Saccharomyces*, *Pichia*, *Aspergillus* and *Malassezia*. We found no evidence of an effect of age on the core gut mycobiome structure. Further studies in different African countries are needed to reach a global view of the core gut mycobiome of African children. They should also aim to compare the digestive mycobiome of breastfed African children with their mother’s breast milk mycobiome.

## Figures and Tables

**Figure 1 microorganisms-12-00926-f001:**
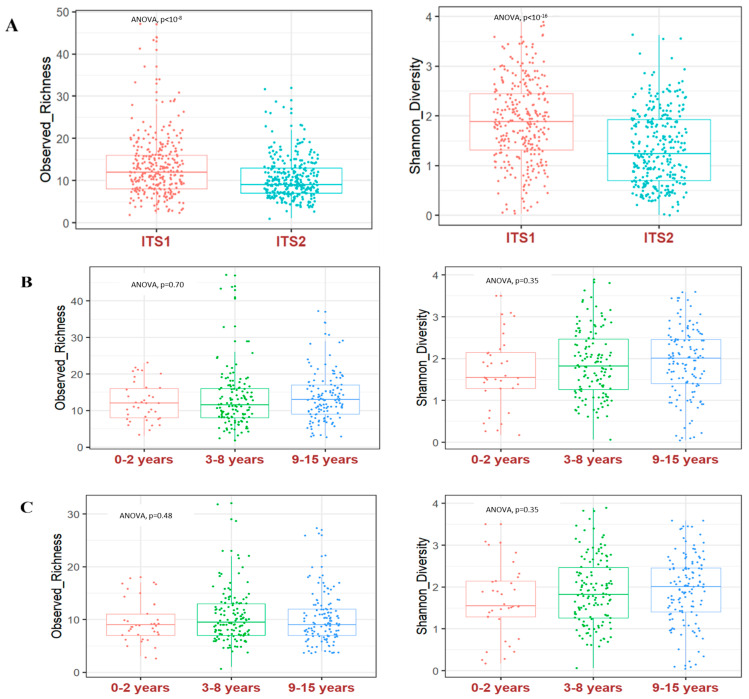
Malian children’s fungal gut community structure. Fungal alpha diversity according to the amplified ITS genomic region and the children’s age group. (**A**) Observed species richness and Shannon diversity obtained by using ITS1 or ITS2. Observed species richness and Shannon diversity obtained by using ITS1 (**B**) or ITS2 (**C**) in comparison with age. Boxplots show medians with first and third quartiles (25% to 75%).

**Figure 2 microorganisms-12-00926-f002:**
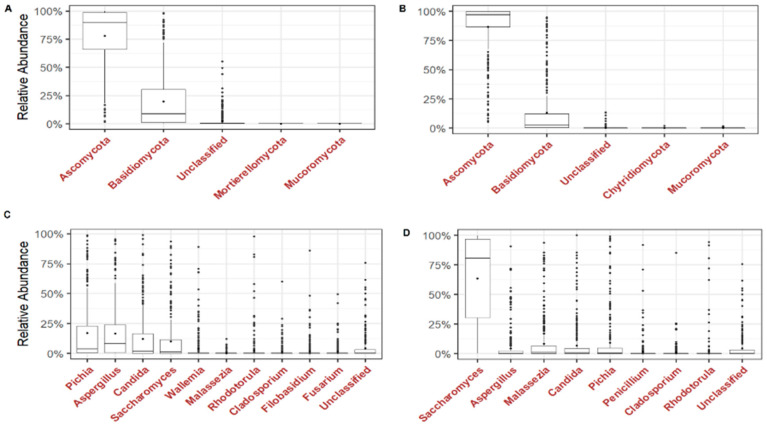
Malian children’s fungal gut community structure. Relative abundance of fungi at the phylum and genus level. Relative abundance of fungal phyla in each sample in ITS1 (**A**) and ITS2 (**B**). Relative abundance of fungal genera in each sample in ITS1 (**C**) and ITS2 (**D**). Boxplots show medians with first and third quartiles (25% to 75%).

**Figure 3 microorganisms-12-00926-f003:**
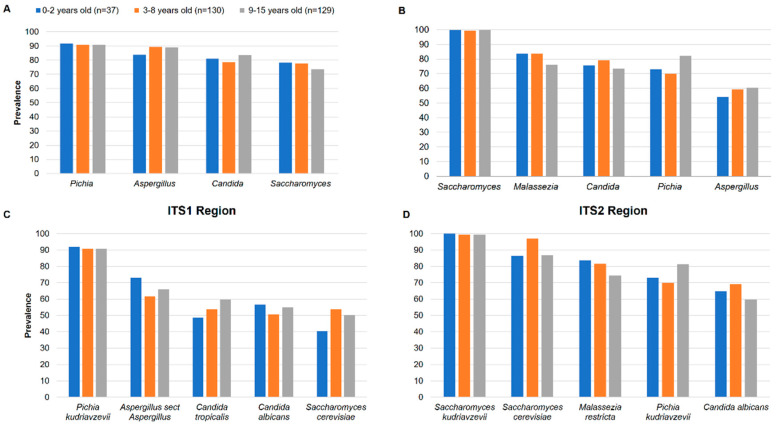
Malian children’s fungal gut community structure. Distribution of fungal genera and species according to age obtained with ITS1 (**A**,**C**) or ITS2 (**B**,**D**) metabarcoding. Only genera and species present in at least 50% of the children are shown.

**Figure 4 microorganisms-12-00926-f004:**
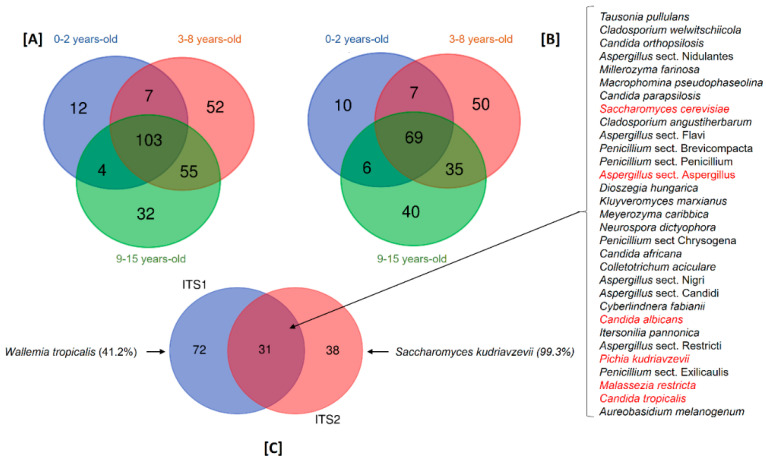
Malian children’s fungal gut community structure. The mycobiome shared by children. Fungal species detected by using ITS1 (**A**) or ITS2 (**B**). (**C**) Mycobiome amplified by one or both regions comparing the common mycobiome from ITS1 and ITS2. Red text indicates the species present in at least 50% of the children.

**Table 1 microorganisms-12-00926-t001:** Malian children’s fungal gut community structure. The prevalence of genera and species present in at least 50% of children with the ITS1 and 2 regions.

Fungal Genera	ITS1 Barcode	ITS2 Barcode
*Pichia*	90.9%	75.7%
*Aspergillus*	88.5%	59.0%
*Candida*	81.1%	76.3%
*Saccharomyces*	76.0%	99.6%
*Malassezia*	<50%	80.4%
**Fungal Species**		
*Pichia kudriavzevii*	90.9%	75.3%
*Aspergillus* sect. *Aspergillus*	66.9%	Not detected
*Candida tropicalis*	55.7%	<50%
*Candida albicans*	53.4%	64.5%
*Saccharomyces cerevisiae*	50.7%	91.2%
*Saccharomyces kudriavzevii*	Not detected	99.3%
*Malassezia restricta*	<50%	78.7%

## Data Availability

The metabarcoding data are available at DOI: 10.13140/RG.2.2.36194.71366 (accessed on 1 March 2024).

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
