# Peer review of "Malian Children’s Core Gut Mycobiome"

_microorganisms, 2024, doi:10.3390/microorganisms12050926_

Round 1
Reviewer 1 Report
Comments and Suggestions for Authors
The aim of this study was to describe the core mycobiome of the gut and to compare the structure of the fungal community in the gut of breastfed Malian children aged 0-2 years with other age groups. The study analyzed rRNA ITS1 and ITS2 metabarcoding data from 296 stool samples from children and found that the core mycobiome of the gut includes Saccharomyces cerevisiae, Candida albicans, Pichia kudriavzevii, Malassezia restricta, Candida tropicalis and Aspergillus. Overall, the manuscript appears to be a well-written and comprehensive study of intestinal mycobiomas in Malian children. The authors have presented a thorough analysis of the predominant fungal taxa, the core intestinal mycobiome, and the differences observed when using the ITS1 and ITS2 barcoding regions. The interpretations are well supported by the data presented and the existing literature, and the authors have highlighted the main strengths and limitations of their study. The conclusions drawn are reasonable and provide a solid foundation for future research on gut mycobiomes in African children. I enjoyed reading this manuscript. There are some revisions that the authors should make before finalizing the manuscript.
Introduction
- - The authors' decision to use the dataset from a study on the influence of microbiota in malaria susceptibility in Malian children to describe the overall gut mycobiome of Malian children is not clearly justified in the introduction.
- - Generalizing the findings from a specific subgroup of Malian children (those susceptible to malaria) to the broader Malian child population may not be straightforward.
- It would be important for the authors to briefly mention the reasons that motivated them to analyze both the ITS1 and ITS2 regions of the fungal rDNA.
Patients and Methods
- - The authors should have mentioned the manufacturer, city, and country of origin for the AmpliTaq Gold master mix, similar to how they provided the details for the primers (Eurogentec, Seraing, Belgium).
- - Specify the number of amplification cycles used in the PCR procedure.
- - The authors mention diluting the amplicons to a concentration of 3.5 ng/µl after purification on AMPure beads. However, they do not provide the rationale for selecting this specific concentration.
- - Specify the version of the Qubit instrument used
- - Line 85-86. This reviewer almost sure that the Illumina sequencing adapters were already incorporated into the primers, and only the dual-index barcodes were added at this stage.
- - Line 93. "fastaq" - it should be "fastq"
- It is unclear if the authors took steps to normalize the sequence counts across samples prior to calculating diversity metrics.
Results
- - The authors should have provided the total number of samples collected earlier in the manuscript “Study Participants and Stool Collection”. This information would give the reader a clear understanding of the study cohort and data availability before reaching into the results.
- - Provide an explanation for the difference between the 300 enrolled children and the 296 children included in the analysis.
- - Line 129. Why the authors decided on these specific age categories instead of, for example, a 2-group or 4-group classification.
- - The column header in table 1, change “% of children harbouring the fungal genus” to “Fungal genera/species”
- Figure 4. The figure should be updated to clearly display the (A), (B), and (C) labels as described in the legend.
Discussion
- - Line 207-208. The authors stated that “This suggests a relatively stable fungal diversity from birth to adolescence”. The authors could further discuss potential factors that may contribute to the observed stability. They could also provide additional insights into the significance of these findings.
- - Discuss the potential functional role and ecological significance of the identified major fungal genera in relation to gut health, immune system development and host-microbe interactions in children.
Comments on the Quality of English Language
Minor editing of English language required
Reviewer 2 Report
Comments and Suggestions for Authors
This generally sound work could additionally gain in importance if the new interesting data on the mycobiota of infants (in comparison with that of other age groups) were considered in the more inclusive context of human micro-, myco- and protozoan biota. This would enable establishing a link to the issue of its relationship with child (and also adult) physical and mental health state. The authors point out that their contribution forms a part of a more general research program. Therefore, paying attention to this important relationship would logically represent a continuation of their research effort. In terms of this next stage, it would make sense to address the subperiods of the earliest human life period (0-2yrs) separately. In an analogy to our knowldege concerning the microbiota, it could be hypothesized that the adult-like mycobiota under study in the work under review gradually forms, e.g., during the very first months under the influence of both maternal and environmental fungal species (it would be of interest to compare breast- and formula-fed babies as well as neonates delivered naturally and those survivng cesarian section). The modern genetic techniques used to analyze the mycobiota and single out its dominant representatives are sufficiently valid scientifically.
Comments on the Quality of English LanguageLingusitically, some minor grammar and usage errors should be eliminated. For instance, this phrase in the abstract is grammatically inadequate: "The impact of fungi in health anddisease show agreat interest in research". Nevertheless, I realize that I'm not a native speaker myself.
